# Chemical language modeling with structured state space sequence models

Rıza Özçelik [1,2], Sarah de Ruiter[1], Emanuele Criscuolo[1] & Francesca Grisoni [1,2] ✉

Generative deep learning is reshaping drug design. Chemical language models (CLMs) – which generate molecules in the form of molecular strings – bear particular promise for this endeavor. Here, we introduce a recent deep learning architecture, termed Structured State Space Sequence (S4) model, into de novo drug design. In addition to its unprecedented performance in various fields, S4 has shown remarkable capabilities to learn the global properties of sequences. This aspect is intriguing in chemical language modeling, where complex molecular properties like bioactivity can 'emerge' from separated portions in the molecular string. This observation gives rise to the following question: Can S4 advance chemical language modeling for de novo design? To provide an answer, we systematically benchmark S4 with state-of-the-art CLMs on an array of drug discovery tasks, such as the identification of bioactive compounds, and the design of drug-like molecules and natural products. S4 shows a superior capacity to learn complex molecular properties, while at the same time exploring diverse scaffolds. Finally, when applied prospectively to kinase inhibition, S4 designs eight of out ten molecules that are predicted as highly active by molecular dynamics simulations. Taken together, these findings advocate for the introduction of S4 into chemical language modeling – uncovering its untapped potential in the molecular sciences.

Designing molecules with desired properties from scratch is a "needle in the haystack" problem. The chemical universe – estimated to comprise up to $10^{60}$ small molecules[1] – remains largely uncharted. Generative deep learning offers unprecedented opportunities to explore the chemical universe in a time- and cost-efficient manner[2], by enabling the production of desirable molecules without the need for hand-crafted design rules. In particular, chemical language models (CLMs) have yielded experimentally-validated bioactive designs[3–7] and stood out as powerful molecular generators[2,8–13].

CLMs adapt algorithms developed for sequence processing to learn the "chemical language", that is, how to generate molecules that are chemically valid (syntax) and possess desired properties (semantics)[7]. This is achieved by representing molecular structures as string notations, such as the Simplified Molecular Input Line Entry Systems (SMILES[14], Fig. 1a), among others[15,16]. These molecular strings are then used for model training and subsequent generation of molecules in textual form. Compared to generative methods based on molecular graphs[17], CLMs can learn more complex molecular properties better[8], and generate increasingly larger molecules more efficiently[18,19]. These aspects have made CLMs become one of the de facto approaches for de novo drug design.

Several CLM architectures have been proposed for de novo design[20], the most popular of which are long short-term memory (LSTM)[3–5,21,22] models. LSTMs are trained to produce molecular strings element-by-element and have fast generation capabilities. However, the iterative structure forces those models to compress the sequence

[1]Institute for Complex Molecular Systems and Department of Biomedical Engineering, Eindhoven University of Technology, Eindhoven, The Netherlands. [2]Centre for Living Technologies, Alliance TU/e, WUR, UU, UMC Utrecht, Utrecht, The Netherlands. ✉e-mail: f.grisoni@tue.nl

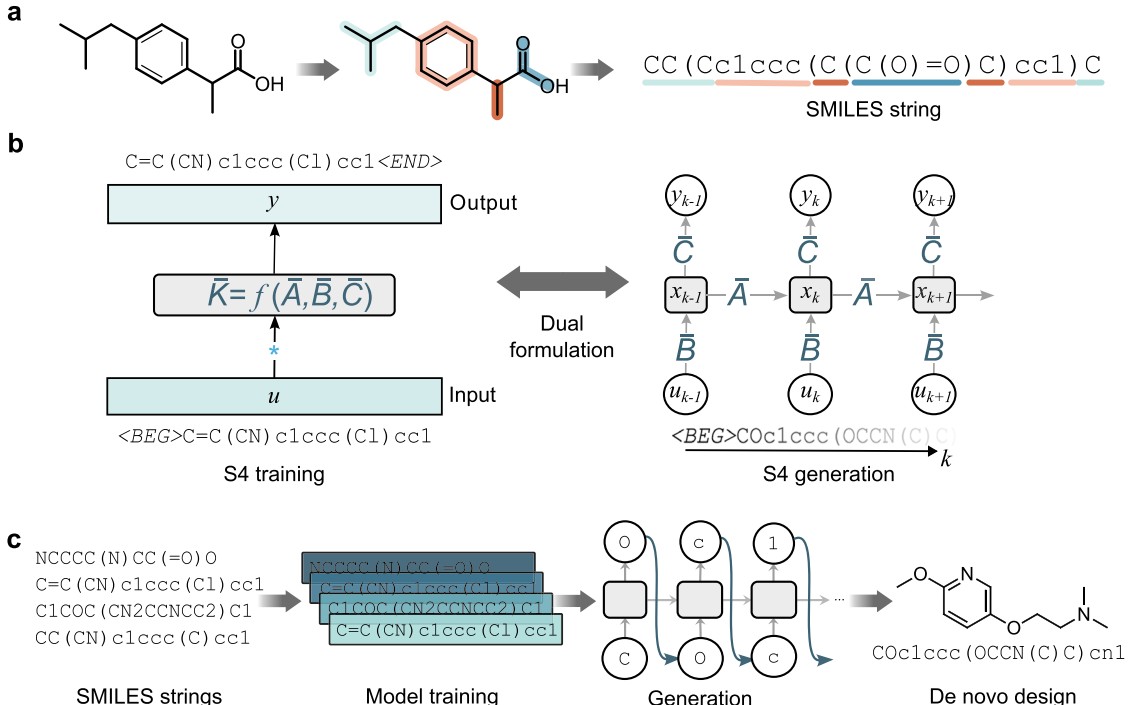

**Fig. 1 | Key concepts of structured state space sequence (S4) models for chemical language modeling. a** Simplified Molecular Input Line Entry System (SMILES) strings[14], used as the chemical language. SMILES strings are obtained by traversing the molecular graph and annotating atom types, rings, and bond types in the form of text. **b** S4 for de novo SMILES design. During training, S4 is formulated as a global convolution and processes the whole molecular string simultaneously. The global convolution filter $\overline{K}$ is parameterized via the matrices $\overline{A}$, $\overline{B}$, and $\overline{C}$ (Eq. (2)). During the generation, S4 switches to the recurrent formulation (via the same parameters, (Eq. (1)) and produces SMILES strings element-by-element for more efficient and effective chemical space exploration. **c** Computational pipeline, where S4 was used to learn from known SMILES strings and generate new molecules de novo.

into an information bottleneck and challenges the learning of global sequence properties[23–25]. Transformers[26], a more recent architecture, overcome this bottleneck by processing the entire input molecular string at once[27,28]. LSTMs and GPTs present different – and somewhat complementary – strengths and weaknesses when it comes to de novo molecule design[25,29–32]. The recurrent nature of LSTMs allows learning local properties better than GPTs, while GPTs capture global properties better thanks to their 'holistic' processing[25]. Moreover, while LSTMs remain efficient, Transformers become increasingly compute-intensive when generating progressively longer SMILES strings, which might limit their broad applicability in the chemical sciences. These aspects make it necessary to stretch the boundaries of current CLM approaches further, to chart the chemical space more effectively in search for bioactive molecules[25].

Structured state space sequence models (S4s) are a recent member of the fast-growing family of state space architectures[33–36], which are gathering increasing attention in the deep learning community[37–40]. S4s showed outstanding performance in audio, image, and text generation[35] and have a "dual nature": they (1) are trained over the entire input sequences to learn complex global properties and (2) generate one string element at a time – thereby combining some respective strengths of Transformers and LSTMs. Motivated by such "best of two worlds" behavior, here we ask the following question: Can S4 advance the current state-of-the-art in chemical language modeling? We find evidence that it can.

Here, we apply S4 to chemical language modeling on SMILES strings and benchmark it on various tasks relevant to drug design – from learning bioactivity to chemical space exploration and natural-product design. Moreover, we further corroborate the promise of S4 via the prospective de novo design of kinase inhibitors, validated using molecular dynamics simulations. Our results show the potential of S4 for chemical language modeling, especially in capturing

bioactivity and complex molecular properties. To the best of our knowledge, this is the first time that state space models have been applied to molecular tasks, and we expect their relevance for chemical language modeling to increase in the future.

## Results and discussion

### Structured state space sequence model (S4)

S4s are an extension of discrete state space models, which are widely adopted in control engineering[41]. Discrete state space models map an input sequence $u$ to an output sequence $y$, through the learnable parameters $\overline{A} \in \mathbb{R}^{N \times N}$, $\overline{B} \in \mathbb{R}^{N \times 1}$, $\overline{C} \in \mathbb{R}^{1 \times N}$, and $\overline{D} \in \mathbb{R}^{1 \times 1}$, as follows:

$$x_k = \overline{A}x_{k-1} + \overline{B}u_k$$
$$y_k = \overline{C}x_k + \overline{D}u_k. \tag{1}$$

In other words, discrete state space models define a "linear recurrence": at any step $k$, the $k$-th element of the input sequence $u_k$ is fed into the model and used to update the hidden state $x_k$ and to generate an output, $y_k$. The matrices $\overline{A}, \overline{B}, \overline{C}$, and $\overline{D}$ control how the input and the hidden state are combined to provide an output (Fig. 1b).

Besides their recurrent formulation, discrete state space models can be formulated as a convolution with the same set of parameters. It can be demonstrated that, by "unrolling" the linear recurrence (Eq. (1)), the output sequence $y$ can be obtained via a learnable convolution over the input sequence $u$:

$$y = u * \overline{K}, \tag{2}$$

where $\overline{K}$ is the convolution filter, parameterized via $\overline{A}$, $\overline{B}$, and $\overline{C}$ (see Supplementary Eqs. (1)–(4) for a detailed derivation). This convolutional representation reveals a key aspect of state space models: they

learn explicitly from the entire sequence (via global convolution) while preserving recurrent generation capabilities (Fig. 1b).

Learning the optimal parameters of a discrete state space system, however, introduces vanishing gradients and numerical instabilities in recurrent and convolutional formulations, respectively. Structured state space sequence models, (S4s)[35] tackle those issues by introducing additional structure to the model parameters (via the so-called high-order polynomial projection operators[33]) and reducing the unstable computations to the stable Cauchy kernel[42] computation (see ref. 35 for more detail). Ablation studies[35] have shown the relevance of the added structure to achieve computational feasibility and performance on long sequences. Moreover, such reduction allows S4 to address numerical instabilities encountered in model training and made S4 state-of-the-art in several generative tasks that require learning long-distance relationships[33–35]. Motivated by its performance in other domains and the potential benefits of its dual structure, here we introduce S4 to the molecular sciences for the first time.

We evaluated S4 for its ability to learn from and generate drug-like molecules and natural products in an array of tasks, and in terms of multiple molecular properties. LSTMs and Generative Pretrained Transformers (GPTs) were used as benchmarks, since they are the de facto approaches in chemical language modeling for de novo design[2,7,8,25]. Furthermore, LSTM (recurrent training and generation) and GPT (holistic training and generation) constitute the ideal benchmarks for S4, due to S4's dual formulation (convolution during training and recurrence during generation), which allows inspecting the effect of each of these aspects on the overall performance. Finally, the prospective de novo design of putative mitogen-activated protein kinase 1 (MAPK1) inhibitors, corroborated by molecular dynamics simulations, was performed to test the potential of S4 in real-world drug discovery scenarios.

## Table 1 | Designing drug-like molecules de novo with S4

| Model | Valid | Unique | Novel |
|---|---|---|---|
| S4 | **99,268 (97%)** | **98,712 (96%)** | **95,552 (93%)** |
| LSTM | 97,151 (95%) | 96,618 (94%) | 82,988 (81%) |
| GPT | 93,580 (91%) | 93,263 (91%) | 91,590 (89%) |

The results of LSTM and GPT models on the same tasks are reported for comparison. Each model was trained on 1.9M SMILES strings from ChEMBL and used to generate 102,400 SMILES strings de novo. The number and percentage of valid, unique, and novel molecular designs are reported. The best value per metric is highlighted in boldface.

### Designing drug-like molecules

S4 was analyzed for its ability to design drug-like small molecules (SMILES length lower than 100 tokens) extracted from ChEMBL database[43], by focusing on its ability to (1) learn the chemical syntax, (2) capture structural features relevant for bioactivity, and (3) designing structurally diverse molecules.

**Learning the SMILES syntax.** All investigated CLMs were trained on 1.9M canonical SMILES strings extracted from ChEMBL v31[43]. The generated strings were evaluated according to their (1) validity, i.e., the number (and frequency) of SMILES corresponding to chemically valid molecules; (2) uniqueness, which captures the number (and frequency) of structurally-unique molecules among the designs; and (3) novelty, corresponding to the number (and frequency) of unique and valid designs that are not included in the training set. A high number of "chemically-valid" designs suggests that the model has learned how to generate plausible molecules, while high uniqueness and novelty values indicate little redundancy among the designs and with the training set, respectively. Although these metrics are vulnerable to trivial baselines[44], they provide insights into a model's capacity to learn the SMILES "syntax".

All CLMs generated more than 91% valid, 91% unique and 81% novel molecules (Table 1). Moreover, their designs approximated the training and test sets in terms of selected molecular properties (i.e., octanol-water partition coefficient[45], quantitative estimate of drug-likeness[46], Bertz complexity[47], and synthetic accessibility[48,49]) with no notable differences among architectures (Supplementary Fig. 1 and Supplementary Table 1). These results agree with the literature on CLMs (e.g., refs. 2,50) and demonstrate the robustness of the model training procedure. S4 designs the most valid, unique, and novel molecules, by generating more novel molecules than the benchmarks (from approximately 4000–12,000 more), and displays a good ability to learn the "chemical syntax" of SMILES strings. The potential of S4 in comparison with existing de novo design approaches was further corroborated on the MOSES benchmark[51], where S4 consistently scored among the top-performing deep learning approaches (Supplementary Table 2).

To shed additional light on the strengths and limitations of S4 in comparison with the benchmarks, we analyzed the sources of invalid molecule generation for all methods in terms of branching and ring errors, erroneous bond assignment, and other (miscellaneous) syntax issues (Fig. 2). Interestingly, each method seems to show different types of errors leading to SMILES invalidity. LSTM struggles the most

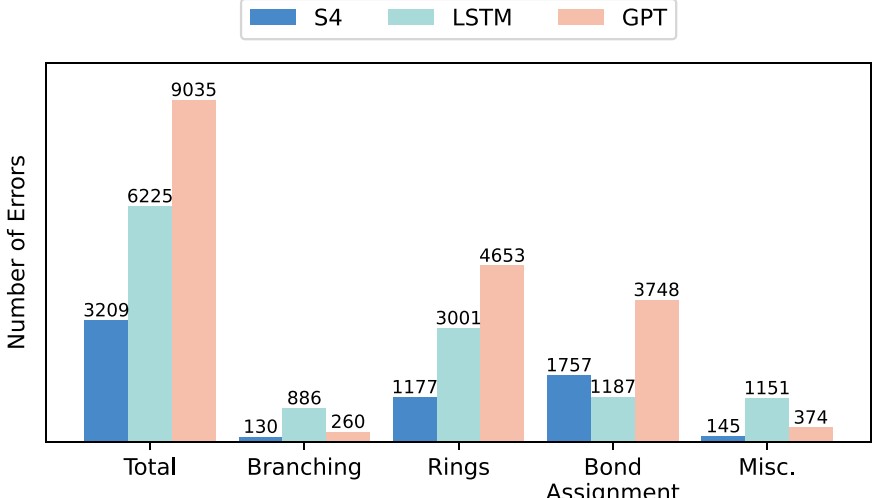

**Fig. 2 | SMILES design errors, grouped by category and CLM architecture.** Each CLM was trained on ChEMBL and used to design 102,400 SMILES strings. The invalid designs were categorized per error, and the reported values indicate the number of errors in each category.

with branching, and performs the best with bond assignment, while GPT struggles the most with rings and bond assignment, and has intermediate performance otherwise. S4 struggles more than LSTM with bond assignment, and generates remarkably fewer errors than both benchmarks in branching and ring design. Our hypothesis is that bond assignment indicates good learning of "short-range" dependencies, while branching and ring opening and closure require better capturing of the "long-range" relationships. This suggests that S4 captures relatively "long-distance" relationships well, in agreement with existing evidence in other domains[33–35].

**Capturing bioactivity.** We evaluated S4 for its ability to learn elements of bioactivity. With CLMs this is often achieved with transfer learning[52], which allows transferring knowledge acquired from one task to another task with fewer available data. Via transfer learning, after pre-training a CLM on a large corpus of SMILES strings, the model can be then "fine-tuned" on a smaller, and task-focused set (e.g., bioactive molecules) by additional training[22]. Here, we performed five fine-tuning campaigns, focusing on distinct macromolecular targets from the LIT-PCBA[53] dataset: (1) pyruvate kinase muscle isoform 2 (PKM2), (2) mitogen-activated protein kinase 1 (MAPK1), (3) glucocerebrosidase (GBA), (4) mechanistic target of rapamycin (mTORC1), and (5) cellular tumor antigen p53 (TP53).

Evaluating the bioactivity of de novo designs (besides synthesis and wet-lab testing) is non-trivial, since this property cannot be fully captured by traditional molecular descriptors, and might not be accurately predicted by quantitative structure-activity relationship models[54,55]. Hence, we used experimentally-tested molecules to evaluate the capacity of a CLM to learn elements of bioactivity retrospectively. Several studies have shown that the likelihoods learned by a CLM during fine-tuning can be used to prioritize designs with high chances of being bioactive[6,56,57]. Based on the same principle, here we used the likelihoods learned by the CLMs to rank existing molecules and evaluate their capacity to prioritize bioactive compounds over inactive ones.

For each of the selected targets, bioactive molecules (Supplementary Table 3) were used for fine-tuning, with ten random training-validation-test splits. After fine-tuning the CLMs on each target, for each training-test split, we proceeded as follows:

(1) With each fine-tuned model and per each target, we predicted the likelihoods (Eq. (4)) of the SMILES strings in the respective test set. The considered test sets resemble a real-world scenario in terms of hit-rate, and they comprise 11 (mTORC) to 56 (PKM2) active molecules and 10,240 inactive molecules (except for TP53, containing 3301 inactive molecules, Supplementary Table 3);

(2) We ranked the molecules of the test set according to the predicted likelihoods (Eq. (5));

(3) For each target and each test set, we computed the fraction of actives ranked among the top 10, top 50, and top 100 molecules. The higher the number of active molecules ranked in early portions of the test set by a CLM, the better the model has learned what is relevant for bioactivity on the investigated target after fine-tuning.

Our results show variable performance depending on the target (Fig. 3). The most challenging target is TP53, on which no model could consistently retrieve actives among the top 10 scoring molecules. Notably, this target has the most challenging test set, where inactive molecules are similar to the actives of both the training and the test sets (Supplementary Fig. 2), potentially indicating the presence of activity cliffs[58]. MAPK1 and mTORC1 also challenge the CLMs; here, S4 retrieved more active molecules than the benchmarks, especially in the early portions of the test set. PKM2 and GBA are the easiest datasets; here, all CLMs identified bioactive molecules in their top 10, with S4 achieving the highest median across the board. A Wilcoxon signed-ranked test[59] on the pooled scores across datasets supports the superior performance of S4 compared to the benchmarks ($p$ [top 10] = 8.41e−6, $p$ [top 50]= 2.93e−7, $p$ [top 100] = 1.45e−7 compared to LSTM, and $p$ [top 10] = 2.33e−3, $p$ [top 50] = 3.72e−3, $p$ [top 100] = 2.61e−2 compared to GPT), and of GPT compared to LSTM ($p$ [top 10] = 5.22e−3, $p$ [top 50] = 3.75e−5, $p$ [top 100] = 2.02e−6).

Under the constraints of the study design, these results indicate that processing the input SMILES "holistically" (as GPT and S4 do) leads to capturing complex properties like bioactivity better, with a better performance obtained by S4.

**Chemical space exploration.** We analyzed the ability of S4 to explore the chemical space, in terms of generating structurally diverse and bioactive molecules. To this end, we employed a commonly-used strategy with CLMs, that is, varying the sampling temperature ($T$) to control chemical diversity[60]. $T$ affects which elements of a string are generated by a weighted random sampling (Eq. (3)). When $T \to 0$ the most likely element (based on the CLM prediction) is selected as the next element of the sequence, while the higher the $T$, the more random the selections. $T = 1$ corresponds to using the CLM predictions as the sampling probability of each element at each generation step.

We experimented with an increasing sampling temperatures (from $T = 1.0$ to $T = 2.0$ with a step of 0.25). Each $T$ value was used to generate 10,240 SMILES strings per model across the five chosen targets and all training-test splits. Then, we evaluated the designs based on three metrics (Fig. 4):

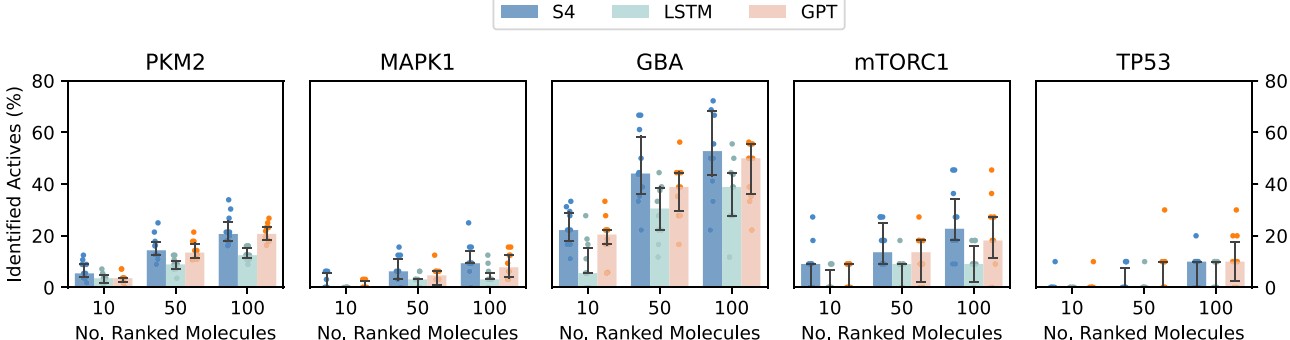

**Fig. 3 | Retrospective enrichment analysis for all models across five selected macromolecular targets.** The fine-tuned models were used to rank the held-out actives and inactives of the respective protein targets. The percentage of known actives ranked per considered number of test set molecules (10, 50, 100) was computed across ten runs. Bar heights report the median across runs and error bars report the first and third quartiles ($n = 10$). Source data are provided as a Source Data file.

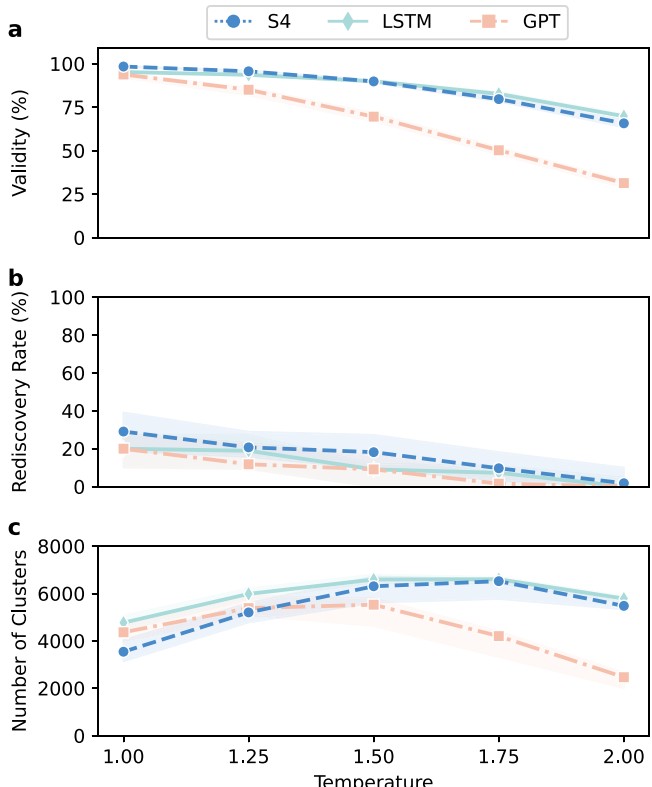

**Fig. 4 | Model performance when varying the temperature value.** Each model was analyzed for its performance when varying the temperature values from 1.0 to 2.0, with a step of 0.25, and by sampling 10,240 molecules. **a** Analysis of the SMILES validity across temperature. **b** Variation of rediscovery rate. The models were evaluated for their capability to rediscover bioactive molecules not used for model training or design molecules similar in structure (i.e., with a Tanimoto similarity on extended connectivity fingerprints higher than 60%). **c** Analysis of the number of diverse groups of scaffolds generated per method. Scaffolds were clustered together if they had a Tanimoto similarity (computed on extended connectivity fingerprints) larger than 60%. For each plot, the solid line indicates the median obtained across the five analyzed protein targets (PKM2, MAPK1, mTORC1, and TP53) with ten runs each ($n = 50$), and the shaded area indicates the inter-quartile range. The statistics per individual target can be found in Supplementary Fig. 3. Source data are provided as a Source Data file.

- The validity of the generated strings, which captures how robust the model is to increasing degrees of randomness in preserving a correct syntax. The higher the validity, the better.
- Rediscovery rate: de novo design models are often evaluated for their capacity to reproduce existing molecules with experimentally verified biological activities[50,55]. For this purpose, we used the held-out actives previously described for each target. Moreover, to "relax" the criterion of rediscovery, we considered held-out actives with substructure similarity higher than 60% to a de novo design (as computed via Tanimoto similarity on extended connectivity fingerprints[61]) to compute rediscovery. Higher rediscovery rates at increased temperature values indicate that the model can explore regions related to bioactivity despite increased randomness.
- Scaffold diversity: designing molecules with novel scaffolds bears relevance in lead identification[62], and can be used as a proxy to evaluate CLMs[51]. Here, to have a better evaluation of what constitutes a novel scaffold, the novel designs were grouped in clusters based on their scaffold similarity. This was achieved via hierarchical clustering, to group designs with similar Bemis-Murcko scaffolds[63] (as computed via the Tanimoto similarity on

the corresponding extended connectivity fingerprints[61] higher than 60%). Only novel and unique scaffolds were considered. We then counted the number of obtained scaffold clusters, the higher, the better.

The models display similar trends with increasing $T$ values for all the analyzed factors across datasets, with varying magnitude (Fig. 4). In general, the validity decreases with increasing temperature (as previously observed[60]), with the highest effect observed for GPT (median validity across training setups getting lower than 40%, Fig. 4a). Both S4 and LSTM show higher robustness than GPT to increasing temperature values (with LSTM performing slightly better for $T \geq 1.75$), suggesting that sequential generation can boost chemical space exploration. S4 outperforms LSTM in terms of rediscovery rate (Fig. 4b), in agreement with our previous results on bioactivity (Fig. 3). We also compute the exact rediscovery rate (identical molecular structure) and observe that no model can consistently generate held-out actives. When it comes to the diversity of the designs (Fig. 4c), LSTM can generate the highest number of structurally unique scaffolds (median across datasets and setups: 6602 clusters, $T = 1.75$) and S4 is the close second-best model (6520 clusters, $T = 1.75$). While GPT obtains a suboptimal performance across the board, LSTM seems better for chemical space exploration when bioactivity is not the main objective, while S4 can better capture bioactivity and preserve a good chemical space exploration at the same time, combining the strengths of the two benchmarks with its dual structure. These results confirm the promise of S4 when it comes to generating structurally diverse and bioactive drug-like molecules.

## Designing natural products
S4 was further tested on more challenging molecular entities than drug-like molecules. To this end, we evaluated its capacity to design natural products (NPs), which are invaluable sources of inspiration for medicinal chemistry[64,65]. Compared to synthetic small molecules, NPs tend to possess more intricate molecular structures and ring systems, as well as a larger fraction of $sp^3$-hybridized carbon atoms and chiral centers[66–68]. These characteristics correspond to longer SMILES sequences on average, with more long-range dependencies, and make natural products a challenging test case for CLMs[19,69].

We trained the CLMs on large natural products (32,360 SMILES strings with length > 100, chosen to complement the previous analysis) from the COlleCtion of Open Natural ProdUcTs (COCONUT) database[70]. We then used the CLMs to design 102,400 SMILES strings de novo and computed the fraction of valid, unique, and novel designs (Table 2). All CLMs can design natural products, with lower performance compared to drug-like molecules. S4 designs the highest number of valid molecules by approximately 6000 to 12,000 molecules (7–13% better), and LSTM achieves the highest novelty by approximately 2000 molecules (2%) over S4.

To further investigate the characteristics of the designs, we computed the natural-product likeness[71], which captures how similar a molecule is to the chemical space covered by natural products in terms of its substructures (the higher the NP-likeness, the more similar). The novel designs of S4 have significantly higher values of NP-likeness than the benchmarks (Mann–Whitney U test, $p = 1.41e{-}53$ compared to LSTM, and $p = 1.02e{-}82$ compared to GPT), closer to the values of the training and test sets on average (Table 2). Moreover, the NP-likeness values better match the distribution of the COCONUT molecules in terms of Kolmogorov–Smirnov (KS) distance[72], which quantifies how much the cumulative distributions of two observations differ (between 0% and 100%; the lower, the closer the distributions).

In addition to NP-likeness, we evaluated the novel designs in terms of several structural properties important for natural products[66–68], namely: the number of $sp^3$-hybridized carbon atoms, aliphatic rings,

**Table 2 | Natural product design with CLMs**

| Metric | | S4 | LSTM | GPT | Training | Test |
|---|---|---|---|---|---|---|
| Syntax | Valid | **82,633 (81%)** | 76,264 (74%) | 70,117 (68%) | n.a. | n.a. |
| | Unique | **53,293 (52%)** | 51,326 (50%) | 50,487 (49%) | n.a. | n.a. |
| | Novel | 40,897 (40%) | **43,245 (42%)** | 43,168 (42%) | n.a. | n.a. |
| NP likeness | Value | **1.6 ± 0.7** | 1.5 ± 0.7 | 1.5 ± 0.7 | 1.6 ± 0.7 | 1.6 ± 0.7 |
| | $KS_{train}$ | **4.03%** | 5.89% | 9.44% | 0.00% | 0.81% |
| | $KS_{test}$ | **4.51%** | 6.60% | 10.13% | 0.81% | 0.00% |
| No. sp³ carbons | Value | 42 ± 16 | 44 ± 17 | 43 ± 16 | 38 ± 16 | 37 ± 15 |
| | $KS_{train}$ | **13.96%** | 17.31% | 14.51% | 0.00% | 1.02% |
| | $KS_{test}$ | **14.08%** | 17.45% | 14.34% | 1.02% | 0.00% |
| No. aliphatic rings | Value | 6 ± 4 | 6 ± 4 | 6 ± 4 | 7 ± 4 | 6 ± 4 |
| | $KS_{train}$ | **5.65%** | 6.91% | 8.12% | 0.00% | 1.08% |
| | $KS_{test}$ | **5.41%** | 6.25% | 7.56% | 1.08% | 0.00% |
| No. spiro atoms | Value | 0.3 ± 0.9 | 0.3 ± 0.8 | 0.3 ± 0.7 | 0.6 ± 1.2 | 0.6 ± 1.2 |
| | $KS_{train}$ | **10.81%** | 12.88% | 12.71% | 0.00% | 0.21% |
| | $KS_{test}$ | **10.87%** | 12.93% | 12.77% | 0.21% | 0.00% |
| Molecular weight | Value | 1114 ± 315 | 1180 ± 360 | 1119 ± 307 | 1061 ± 295 | 1063 ± 290 |
| | $KS_{train}$ | **9.23%** | 16.97% | 11.02% | 0.00% | 1.40% |
| | $KS_{test}$ | **9.04%** | 16.67% | 10.75% | 1.40% | 0.00% |
| Size of the fused ring system | Value | 5 ± 2 | 5 ± 2 | 5 ± 2 | 5 ± 2 | 5 ± 2 |
| | $KS_{train}$ | **8.05%** | 9.42% | 11.19% | 0.00% | 0.60% |
| | $KS_{test}$ | **7.93%** | 9.44% | 11.21% | 0.60% | 0.00% |
| No. heavy atoms | Value | 78 ± 22 | 83 ± 25 | 79 ± 21 | 75 ± 20 | 75 ± 20 |
| | $KS_{train}$ | **7.76%** | 15.81% | 9.73 | 0.00% | 1.24% |
| | $KS_{test}$ | **7.30%** | 15.34% | 9.31 | 1.24% | 0.00% |

The models were trained on 32,360 natural product SMILES strings from the COCONUT database[70] and used to generate 102,400 SMILES strings de novo. The number and fraction of valid, unique, and novel molecular designs are calculated for each model. For the designs, the mean and standard deviation of (1) the natural-product-likeness values[71], (2) the number of sp³ carbons, (3) the number of aliphatic rings, (4) the number of spiro atoms, (5) molecular weight, (6) size of the largest fused ring system, (7) the number of heavy atoms and the corresponding Kolmogorov–Smirnov distance to the training and test sets ($KS_{train}$ and $KS_{test}$, respectively) are reported. The same statistics from train and test sets (32,360 and 5000 natural products, respectively) are reported for comparison. For each CLM and each metric, the best value is highlighted in boldface. All descriptors were computed on valid, unique, and novel SMILES.

spiro atoms and heavy atoms, as well as the molecular weight and the size of the largest fused ring system. These properties provide additional evidence on the molecular characteristics of the designs, and their structural complexity in comparison with the training natural products. Here, S4 achieved the lowest KS distance to the training and test sets across properties, indicating that its designs match the training natural products best. These results confirm the ability of S4 to learn complex molecular properties for de novo design.

Finally, we analyzed the training and generation speed of the CLM architectures when increasing the SMILES length, to test their practical applicability when designing bigger molecules, like natural products. Our analysis highlighted that S4 is as fast as GPT during training (both are approximately 1.3 times faster than LSTM), and the fastest in terms of generation (Supplementary Fig. 4), thanks to its dual formulation. This further advocates for the introduction of S4 as an efficient approach for molecule design, that "makes the best of both worlds" compared to GPT and LSTM.

**Prospective de novo design**

We conducted a prospective in silico study with S4, focused on designing inhibitors of mitogen-activated protein kinase 1 (MAPK1), a relevant target for oncological therapies[73]. The putative bioactivity of the designs was then evaluated via molecular dynamics (MD).

The S4 model previously pre-trained on ChEMBL was fine-tuned with the SMILES strings of 68 manually-curated inhibitors from ChEMBL, having an experimental constant of inhibition ($K_i$) lower than 1 μM on MAPK1. The last five epochs of the fine-tuned model were then used to generate 256K molecules (51,200 designs per each $T$ value, ranging from 1.0 to 2.0 with a step of 0.25).

The designs were ranked and filtered via log-likelihood score (Eq. (5)) and scaffold similarity to the training set (see "Materials and methods" for further details). The ten top-scoring molecules (**1**–**10**, Fig. 5a and Table 3) were considered for further characterization using MD simulations. As a reference for evaluation, we performed MD simulations also for the closest fine-tuning neighbor of the considered designs (compounds **11**–**16**, selected based on scaffold similarity; Fig. 5a and Table 3). The absolute protein-ligand binding free energy (expressed as Δ$G$ – the lower the stronger the predicted binding) for molecules **1**–**16** was computed via Umbrella Sampling[74] (Table 3). The computed Δ$G$ values for known bioactive molecules (**11**–**16**) have a good correspondence with experimental $K_i$ values from ChEMBL (Table 3), confirming the validity of the chosen MD protocol.

Eight out of ten designs (except **1** and **5**) showed a high predicted affinity (Table 3), with Δ$G$ values ranging from Δ$G = -10.3 ± 0.6$ kcal mol$^{-1}$ (**7**) to Δ$G = -23 ± 4$ kcal mol$^{-1}$ (**2**). Interestingly, these affinities are comparable or even surpass those of the closest active neighbor (Δ$G = -9.1 ± 0.8$ kcal mol$^{-1}$ to Δ$G = -13 ± 2$ kcal mol$^{-1}$). The global substructure similarity (measured on extended connectivity fingerprints) of the designs to their closest neighbor ranges from 31% (**10**) to 87% (**4**, Table 3).

The most potent design according to MD predictions is molecule **2** (Δ$G = -23 ± 4$ kcal mol$^{-1}$, Table 3). This molecule – which is the largest one among the designs (Fig. 5a) – engages extensively with the binding pocket of MAPK1 (Fig. 5b), which explains the remarkably favorable predicted affinity. Design **2** has a limited substructure similarity to its closest bioactive neighbor (molecule **12**, similarity equal to 57%); however, its synthetic accessibility may be limited. Design **3** is

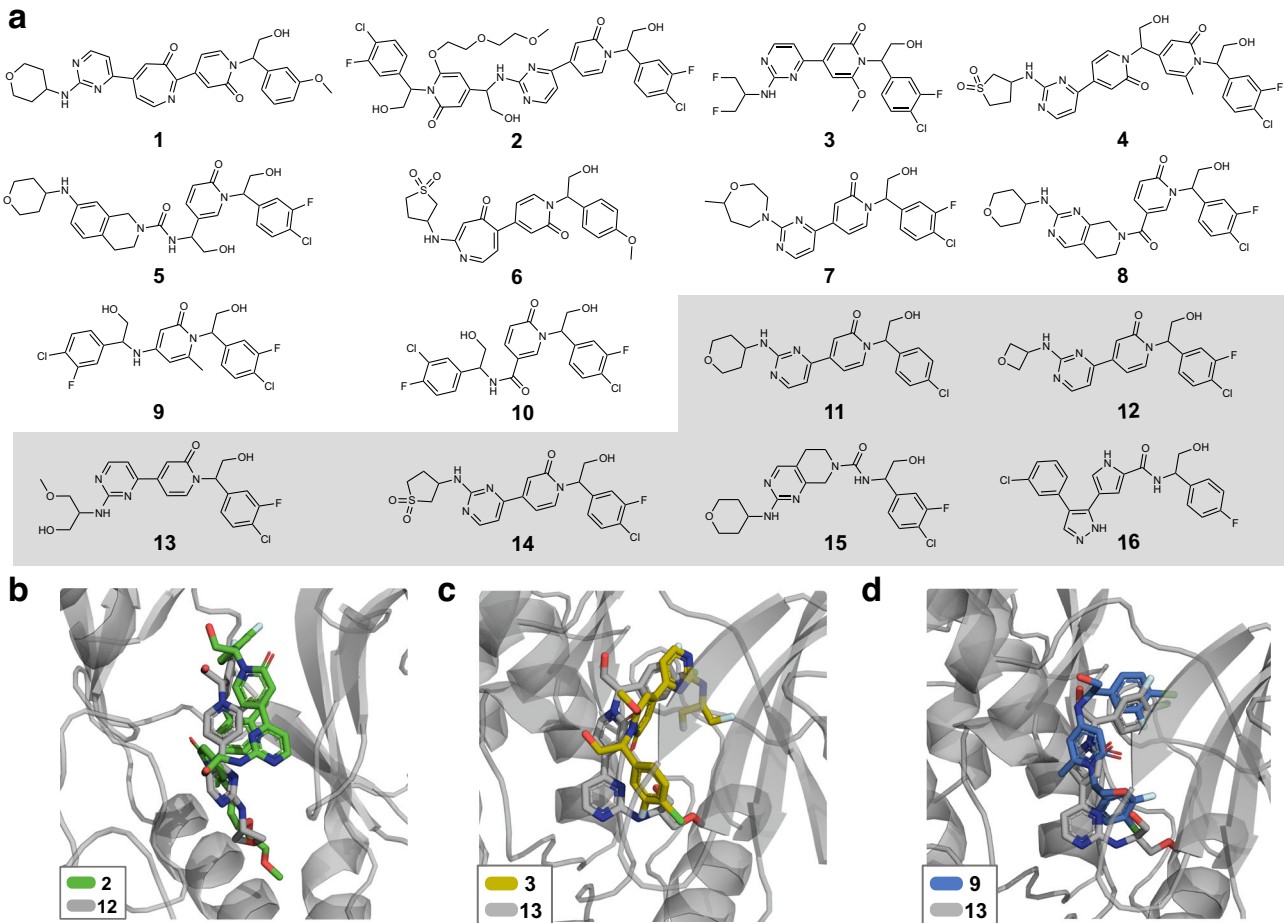

**Fig. 5 | Prospective de novo design of putative MAPK1 inhibitors with S4.**
**a** Selected de novo designs (molecules **1**–**10**) for further characterization. For each de novo design, its most similar training MAPK1 inhibitor (as reported in Table 3) is depicted (compounds **11**–**16**, gray box). The ligand binding pose (obtained via Umbrella Sampling) of selected designs interacting with MAPK1 (PDB-ID = 2Y9Q), in comparison with their most similar bioactive molecule from the fine-tuning set is also depicted: **b** Design **2** (green) compared with compound **12** (gray). **c** Design **3** (yellow) compared with compound **13** (gray). **d** Design **9** (blue) compared with compound **13** (gray).

## Table 3 | In silico prospective study on designing mitogen-activated protein kinase (MAPK1) inhibitors with S4

| S4 design | | Most similar training active | | | Scaffold Similarity | Global Similarity |
|---|---|---|---|---|---|---|
| ID | $\Delta G$ [kcal mol$^{-1}$] | ID | $\Delta G$ [kcal mol$^{-1}$] | $K_i$ [nM] | | |
| **1** | −5.6 ± 0.9 | **11** | −9.1 ± 0.8 | 0.1 | 79% | 65% |
| **2** | **−23 ± 4** | **12** | −12 ± 2 | 0.4 | 63% | 57% |
| **3** | **−19.6 ± 0.9** | **13** | −10.5 ± 0.7 | 3.0 | 100% | 65% |
| **4** | **−13 ± 2** | **14** | −11 ± 3 | 2.5 | 90% | 87% |
| **5** | −7 ± 2 | **15** | −13 ± 2 | 0.6 | 73% | 85% |
| **6** | −11 ± | **14** | −11 ± 3 | 2.5 | 56% | 56% |
| **7** | −10.3 ± 0.6 | **11** | −9.1 ± 0.8 | 0.1 | 50% | 52% |
| **8** | −11.2 ± 0.4 | **15** | −13 ± 2 | 0.6 | 58% | 72% |
| **9** | **−17 ± 2** | **13** | −10.5 ± 0.7 | 3.0 | 41% | 42% |
| **10** | **−15 ± 2** | **16** | −9.1 ± 0.2 | 63.0 | 30% | 31% |

The absolute binding free energy of interaction ($\Delta G$, the lower, the better) was determined via Umbrella Sampling. Values are reported as the average over three repeats, along with the corresponding standard deviation. De novo designs **1**–**10** are compared with the closest inhibitor from the fine-tuning set, selected based on the scaffold similarity (Tanimoto coefficient on extended connectivity fingerprints [ECFPs], using Bemis-Murcko scaffolds[63]). The experimentally-determined inhibition of MAPK1 for compounds **11**–**16** (constant of inhibition, $K_i$) are also reported. Designs whose predicted $\Delta G$ is remarkably better than that of the closest bioactive neighbor are highlighted in boldface. Global substructure similarity to the closest neighbor (as measured by the Tanimoto coefficient on ECFPs) is reported.

predicted with the second highest affinity ($\Delta G = -19.6 \pm 0.9$ kcal mol$^{-1}$), and it shares the same scaffold of compound **13**. Design **3** differs from **13** by the replacement of the ether and hydroxy moieties with two fluorine atoms, and the addition of a methoxy group (Fig. 5a, global similarity equal to 65%). Interestingly, this structural modification leads to an improvement of the predicted $\Delta G$ value (of approximately −10 kcal mol$^{-1}$), possibly due to the ability to penetrate deeply into the binding pocket thanks to the fluorine atoms (Fig. 5c). Halogens are, in fact, favorable for MAPK1, as evident from the fine-tuning molecules (91% of them containing at least one halogen) and existing literature (e.g., refs. 75–78). Evidence of a favorable positioning of halogens is shown on both the "top"[75,76] and "bottom"[77,78] of the binding pocket, further supporting the predicted affinity of compound **3**.

Design **9** ($\Delta G = -17 \pm 2$ kcal mol$^{-1}$) features halogens on both sides, unlike its closest neighbor, molecule **13** ($\Delta G = -10.5 \pm 0.7$ kcal mol$^{-1}$, global similarity equal to 33%), from which it also differs in the moiety attached to the pyridonic ring (Fig. 5a). When inspecting the predicted binding pose, it can be observed that the aromatic ring with halogen substituents, hydroxyl, and carbonyl of pyridone are situated in the same region of the binding groove (Fig. 5d). The difference in $\Delta G$ values (approximately 6.5 kcal mol$^{-1}$ in favor of design **9**) could be ascribed, like with molecule **3**, to the presence of halogens in the lower binding pocket region. This might also explain the high predicted affinity of design **10** ($\Delta G = -15 \pm 2$ kcal mol$^{-1}$) – which differs from **9** by a carbonyl and a methyl group.

With 8 out of 10 designs predicted as bioactive on the intended target by MD, with comparable or higher predicted affinities than their closest fine-tuning molecules, these results further corroborate the potential of S4 for de novo drug design.

## Opportunities for molecular S4

In conclusion, this study pioneered the introduction of state space models into chemical language modeling, with a focus on structured state spaces (S4s). The unique dual nature of S4s, involving convolution during training and recurrent generation, makes them particularly intriguing for de novo design starting from SMILES strings.

Our systematic comparison with GPT and LSTM on a variety of drug discovery tasks revealed S4's strengths: while recurrent generation (LSTM and S4) is superior in learning the chemical syntax and exploring diverse scaffolds, learning holistically on the entire SMILES sequence (GPT and S4) excels in capturing certain complex properties, like bioactivity. S4 with its dual nature, makes the best of both worlds": it demonstrated comparable or better performance than LSTM in designing valid and diverse molecules, and systematically outperformed both benchmarks in capturing complex molecular properties – all while maintaining computational efficiency.

The application of S4 to MAPK1 inhibition, validated by MD simulations, further showcases its potential to design potent bioactive molecules. In the future, we will apply S4 prospectively in combination with wet-lab experiments to enhance its impact in the field. Strategies to increase the structural diversity of the considered designs, such as SMILES augmentation[79] and improved ranking protocols, could further boost its potential in medicinal chemistry.

Several aspects of S4 await to be explored in the molecular sciences, such as its potential with longer sequences (e.g., macrocyclic peptides and protein sequences) and on additional molecular tasks (e.g., organic reaction planning[80] and structure-based drug design[81]).

In the future, we envision the relevance of S4 for molecule discovery to increase, and to potentially replace widely established chemical language models like LSTM and GPT. We believe that the provided open-access code will contribute to the adoption and expansion of S4, to further stretch the boundaries of chemical language modeling.

## Methods
### Designing drug-like molecules
**Data curation.** The pre-training set was generated starting from ChEMBL v31[43]. Fine-tuning datasets were extracted from LIT-PCBA[53]. All sets were generated by (1) retaining molecules containing selected atoms (C, H, O, N, S, P, F, Cl, Br, and I), (2) removing salts and disconnected structures, as well as stereochemistry annotations and charge, (3) retaining molecules whose canonicalized SMILES strings contained 100 tokens or fewer. After sanitization, canonicalization, label encoding, and padding (to 100), molecules were randomly split into training, validation, and test sets. For ChEMBL, this led to a training set of 1,900,000, and a validation and a test set of 100,000 and 23,680 molecules, respectively. The number of compounds for each fine-tuning campaign is reported in Supplementary Table 3.

### Training
**Pretraining.** The hyper-parameters of the LSTM and GPT were tuned with random search for 5 days on a single NVIDIA A100 40GB GPU. The defined hyper-parameter space is based on previous work[27,57,60,82] (Supplementary Table 4). 40 LSTM and 35 GPT models were optimized within a 5-day limit. Hyper-parameter search was conducted to maximize the validity during pre-training.

To account for the lack of previous information on optimal hyper-parameters for molecule generation with S4, we implemented a two-step procedure for hyper-parameter tuning. First, 242 models were trained to prioritize hyper-parameters (see Supplementary Table 4).

High-performing hyper-parameter values in terms of validation accuracy were advanced to the second phase, where 108 experiments were conducted. Hyper-parameter search was conducted for 10 days on multiple NVIDIA A100 40GB GPUs to maximize the validity during pre-training.

**Fine-tuning.** Five fine-tuning campaigns were conducted on five targets: PKM2, MAPK1, GBA, mTORC1, and TP53. For each target, ten runs with different training (80%), validation (10%), and test (10%) splits were performed (except for PKM2 where we used 70%–15%–15% due to limited data). Early stopping on the validation cross-entropy was adopted with a patience of five epochs and a tolerance of $10^{-5}$.

**Temperature sampling.** The sampling probability ($p$) of each $i$-th element at any step of the sequence was computed as follows:

$$p_i = \frac{e^{(y_i/T)}}{\sum_j e^{(y_j/T)}} \tag{3}$$

where $y_i$ is the predicted probability of the $i$th element, $T$ is the sampling temperature, and $j$ runs over all tokens in the vocabulary.

**Molecule ranking with log-likelihoods.** The molecules were ranked based on the joint likelihood of the tokens (i.e., SMILES characters) they contain[82]. For each test molecule, the joint log-likelihood ($\mathcal{L}$) by a model ($\mathbf{M}$) was computed as:

$$\mathcal{L}(\mathbf{M}) = \sum_i \log p(t_i) \tag{4}$$

where $t_i$ is the $i$th token of the SMILES string of a given test molecule and $p(t_i)$ is the probability of that token as predicted by the model $\mathbf{M}$; $i$ runs over all the elements in the molecular string.

To only consider the fine-tuning information and remove potential pre-training bias (as previously observed[82]), the pre-training log-likelihood was subtracted from the fine-tuning likelihood, to obtain a final score:

$$\mathcal{L}_{\text{score}}(\mathbf{M}) = \mathcal{L}(\mathbf{M_{ft}}) - \mathcal{L}(\mathbf{M_{pt}}) \tag{5}$$

where $\mathbf{M_{ft}}$ is the fine-tuned model and $\mathbf{M_{pt}}$ is the pre-trained model. The obtained $\mathcal{L}_{\text{score}}$ was used to rank each test molecule, the higher the $\mathcal{L}_{\text{score}}$, the better the rank.

### Natural product design
The COlleCtion of Open Natural ProdUcTs (COCONUT)[70] database was used for model training. Salts, disconnected structures, stereochemistry, and charge annotations were removed. Molecules with canonical SMILES strings longer than 100 characters were used to train the models. A random search strategy was adopted to tune the hyper-parameters of all models, as previously explained. The models were given a 5-day limit on a cloud NVIDIA A100 GPU and 1024 strings were generated by each model. The models with the highest SMILES validity were selected for further evaluation.

### Prospective de novo design
**Data curation.** Fine-tuning data were collected from ChEMBL v33[43]. All annotations for MAPK1 were retained (target ID: CHEMBL4040). Available assay descriptions were manually inspected and analyzed. Molecules whose inhibitory constant ($K_i$) was lower than 1 μM on reliable inhibition assays (CHEMBL3412886, CHEMBL917079) were retained. SMILES canonicalization and removal of stereochemistry and duplicates led to a set of 68 unique SMILES strings for fine-tuning (SMILES strings available in the dedicated GitHub repository).

**Model fine-tuning and de novo design.** The fine-tuning dataset was split into ten train and validation splits (80–20%) to find the optimal number of fine-tuning epochs. Early stopping on validation loss was used, with patience of five epochs and tolerance of $10^{-5}$. The experiments suggested 45 epochs to be optimal; the pre-trained model was fine-tuned on the whole dataset accordingly.

The models of the last five fine-tuning epochs were used to design molecules. A total of 10,240 designs for temperature values ranging from 1.0 to 2.0 (step size 0.25) were generated per temperature and model, totaling $5 \times 5 \times 10,240 = 256K$ designs. The novel and unique molecules among those designs were ranked by their fine-tuning log-likelihood (Eq. (4)) and the top 5000 molecules were selected for further analysis.

The 5000 top-scoring molecules were divided into two groups, based on their similarity to the fine-tuning set. The similarity was measured via Tanimoto similarity on the extended connectivity fingerprints[61] of the Bemis-Murcko scaffolds[63] (using a radius of 3 bonds and 2048 bits), and a threshold of 60% similarity. The designs in the lists were grouped by their most similar training molecule and ranked by the log-likelihood score (Eq. (5)). The highest-scoring molecule in each group was picked. The top five molecules of the design lists (i.e., **1**–**5** for higher similarity, and **6**–**10** for lower similarity) and their most similar actives (**11**–**16**, based on scaffold similarity, Table 3) were selected for molecular dynamics simulations.

**Molecular dynamics simulation.** The protein structure of MAPK1 was sourced from the Protein Data Bank under the accession code 2Y9Q, characterized by a Resolution of 1.55Å and an R-Value Free of 0.177. Initial complex structures resulted from Docking simulations using Vina[83], establishing the binding pose for subsequent investigation via Umbrella Sampling. The setup of the simulation system was facilitated by the CHARMM-GUI web-based graphical interface[84]. A cubic water box with an edge distance of 13Å encapsulated the system, supplemented by a 0.15 M ionic NaCl solution for solvation neutralization. Gromacs software version 2021[85], operationalized on the Dutch supercomputer Snellius, facilitated all simulations. The energy minimization of solvated systems involved a sequence of steps utilizing the steepest descent method and the conjugate gradient algorithm. Subsequently, equilibration occurred through 5 ns NPT (constant Number of atoms, Pressure, Temperature) ensemble after the first 1 ns NVT (constant Number of atoms, Volume, Temperature) ensemble.

**Binding free energy calculation.** The last conformations of the equilibration phase were used as the starting structures of ligand unbinding simulation. The distance-based Steered MD simulation (center-of-mass-pulling method) was used to pull the ligand away from the protein by approximately 30 Å over the course of 4 ns by using a 1000 kJ/(mol nm²) force along the reaction coordinate ($\xi$), with a pulling speed ($v$) set at 0.001 nm/ps. Snapshot intervals of 10 ps generated 400 configurations from these pulling simulations. Different ligands prompted the extraction of varying conformations, ranging from 22 to 28, along the reaction coordinate ($\xi$) at approximately 0.1 nm intervals. These distinct configurations were then employed as the initial points for individual Umbrella Sampling simulations, differing in quantity depending on the specific ligand under study. Each conformation underwent independent NPT equilibration for 5 ns, followed by a 20 ns MD run in triplicate for each ligand. The potential mean force (PMF) was determined via the weighted histogram analysis method (WHAM)[86], a component of Gromacs. The resultant PMF graphs depicted force in kcal mol$^{-1}$, representing the force required to dissociate the ligand from the binding pocket, against the corresponding distance. The computation of the binding free energy ($\Delta G$) for each ligand involved comparing the plateau region of the PMF curve to the energy minimum obtained from each simulation. In total, the Umbrella Sampling simulations spanned 400 to 560 ns,

comprising 3 replicates, thereby accumulating simulation times ranging from 1.2 μs to 1.6 μs for each ligand.

### Software and code
Data preprocessing, scaffold determination, and molecular fingerprint and descriptor calculation were performed with default settings (unless otherwise noted), using RDKit v2020.09.01 in a Python environment. LSTM and GPT were implemented in Keras v2.7.0 (Tensorflow v2.7.1). The S4 code was extracted from the existing Pytorch-lightning v1.15.0 implementation[35] and simplified to rely solely on Pytorch v1.13.1.

### Reporting summary
Further information on research design is available in the Nature Portfolio Reporting Summary linked to this article.

## Data availability
The data used in our study are available on GitHub at the following URL: https://github.com/molML/s4-for-de-novo-drug-design. Source data are provided with this paper. The molecule designs and source data are also available at: 10.5281/zenodo.12666371.

## Code availability
The Python code to replicate and extend our study is available on GitHub at the following URL: https://github.com/molML/s4-for-de-novo-drug-design. The code at the time of publishing is available at: 10.5281/zenodo.12666371[87].

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

## Acknowledgements

This research was co-funded by the European Union (ERC, ReMINDER, 101077879). Views and opinions expressed are however those of the author(s) only and do not necessarily reflect those of the European Union or the European Research Council. Neither the European Union nor the granting authority can be held responsible for them. The authors also acknowledge support from the Irene Curie Fellowship, the Centre for Living Technologies, and SURF (NWO grant EINF-5406). The authors thank Selen Parlar and the Molecular Machine Learning team (H. Brinkmann, C. Izquierdo-Lozano, M. Reksoprodjo, L. Rossen, Y.G. Nana Teukam, D. van Tilborg, L. van Weesep) for their feedback on the manuscript.

## Author contributions

Conceptualization: R.Ö. and F.G. Data curation: R.Ö. Formal analysis: all authors. Investigation: all authors. Methodology: all authors. Software: R.Ö. and S.d.R. Visualization: R.Ö., F.G. and E.C. Writing – original draft: R.Ö. and F.G. Writing – review and editing: all authors.

## Competing interests

The authors declare no competing interests.
