## [Peer Review File · Nature Communications]

Reviewers' Comments:

Reviewer #1 (Remarks to the Author):

Authors have developed a "Structured State-Space Sequence Models for De Novo Drug Design." However, the Manuscript Authors should add the following points to the Manuscript. The Manuscript started with the "Structured State-Space Sequenc"e Model (S4 model.). The authors have chosen the S4 model for drug design. However, It needs to understand the S4 model. Why have they chosen the model? Any special reason? Therefore, the authors should discuss the S4 model more clearly. In the discussion section, authors should discuss the conventional method of drug discovery. The S4 model is also better than the conventional model. The overall workflow or methodology should be depicted for a clearer view of the readers.

Reviewer #2 (Remarks to the Author):

This manuscript proposes an application of Structured State-Space Sequence (S4) generative models to de novo molecule design. The research aim is significant, and this is the first application of S4 to de novo drug design as claimed (as far as I'm aware); however, several critical issues exist regarding how the computational method is technically explained, the performance is evaluated, the results are discussed, and the source code and trained models are shared with the research community. Unfortunately, this study is far from being a complete work. I'm sorry to state this, but the manuscript strongly feels like it was rushed for publication before it was finished (maybe to be the first to apply S4 to de novo drug design). Below, I list my specific concerns/issues:

1)

The literature review is not sufficiently detailed regarding generative AI applications in de novo drug design. Generative AI for small molecule and protein design has become highly popular lately. Also, deep learning architectures such as graph neural networks, GANs and VAEs have widely been utilised for this purpose. These studies should be cited and mentioned, especially in the context of deep learning architectures and produced output.

2)

"2.2 S4 for de novo molecule design"

These tests are not sufficient as a state-of-the-art/baseline comparison. First of all, only validity, uniqueness and novelty are used as performance metrics, which only presents a limited perspective. There are numerous additional metrics to evaluate models from other angles. These metrics should be utilised for evaluation. Second, the authors only compared their model with LSTM and GPT, which are trained by the authors as their own baselines. However, there are many well-known methods/models in the literature for de novo molecule generation, including both CLMs and graph learning-based models. These models should be employed for performance comparison. Finally, the performance should be discussed with regard to the architecture-, data- and modelling-related differences between the S4 model and the others.

3)

"2.3 S4 for capturing bioactivity"

It is not clear how this procedure was handled to accommodate the fine-tuning procedure in the context of transfer learning and the production of likelihoods for the test molecules. Technical

methodology-related details should be provided. Also, a similar problem exists here (as in issue number 2). Comparison with LSTM and GPT is not sufficient.

4)

"Figure 2 and Figure 3"

While the results provided here seem partially promising, they are far from being conclusive. First of all, baselines performed better in some of the targets. There is no discussion at all regarding this. Additionally, some of these results are contradicting. S4 performed the best for mTORC1 in Figure 2, it performed clearly worse compared to LSTM for mTORC1 in Figure 3; why? Please discuss your results...

5)

"On GBA and MTORC1, S4 outperforms GPT by 5 to 9 folds enrichment, and LSTMs by more than 17 folds."

I cannot see this much difference in Figure 2. Maybe I'm missing something; additional explanation is required.

6)

"Interestingly, unlike with bioactivity, in this case LSTM outperforms GPT, on both the validity and the structural diversity with increasing the temperature values (Figure 4a,b). S4 shows an intermediate behavior between LSTM and GPT. "

This sounds like clever wording, not to admit that LSTM clearly outperformed S4 in terms of the model performance in terms of generation performance with varying temperature values (Figure 4a and b). Please, do not go around the obvious; just say it. Again, there is no discussion at all.

7)

"2.5 Generation speed of S4"

Again, the fact that LSTM outperformed S4 is ignored here. By looking at the results provided in this manuscript, I think the overall discussion (considering all test results provided in 2.2, 2.3, 2.4 and 2.5) should be that LSTM performed better than (or at least nearly as good as) S4 in many of the tests (except maybe some of the results in Figure 2 and 3), which indicates LSTM, an old but effective architecture, is as promising as the S4 model in de novo molecule design until further analyses indicate otherwise.

8)

It is strange that not one de novo molecule is shown in a study in which a de novo molecule design model is proposed. Sample molecules from each test should be given, and models should be compared to each other over the generated molecules. This can be shaped in the context of a use case study, in which a few example de novo molecules generated by the S4 model are qualitatively compared with the output of other models from physicochemical and functional perspectives.

9)

One of the most critical issues is the missing technical explanation of the architecture in the methods section. Authors correctly claim that S4 models are first used here for de novo molecule generation, but they did not explain the details of the architecture. I'm aware that the authors did not propose the S4 architecture itself; it has been previously developed by others, but a thorough technical explanation is required in a study where a rather new architecture is applied in another field for the first time.

10)

An ablation study is required to evaluate the contribution of different modules of the proposed S4-based model to the produced output. This could be done by eliminating certain modules or replacing them with more conventional counterparts. The change in the results should be discussed in relation to the original design.

11)

The authors just stated that the code will be shared over the given GitHub repo link (the repo is empty as of the last week of October, 2023). In today's research, open science is one of the most important aspects, and sharing of the scripts/source code is an important part of it. The source code, datasets and results should be properly shared before the peer review process, not after (so that the reproducibility could be checked). On top of sharing the source code, please prepare a detailed readme file that includes information regarding how your implementation can be used to reproduce the results provided in the manuscript. In addition, please provide information on how the implementation can be used to analyze independent datasets belonging to the user. This way, future researchers can use your method in their own research, and they also can replicate your results when and where necessary.

Reviewer #3 (Remarks to the Author):

This manuscript by Riza et al. introduces a new generative model for de novo generation of molecules from the control engineering research. This is the first time to apply the structured state-space sequence (S4) model (existing approach, SOAT in other domains) to drug design. Authors benchmarked the methods with the current two SOTA methods (LSTM and GPT) on various small molecule drug discovery-related tasks and showed it outperforms all the other methods. The methodology is described clearly, and the code in GitHub is well-documented and maintained.

However, I have serious reservations regarding the value of the proposed approach for drug discovery, especially on small molecule drug design, based on the method and results.

1. The most important issue is the motivation for applying S4 to SMILES-based small molecule generation. S4 was proposed to deal with very long sequences, more than 10000 or more steps, and it has been proven to be effective in many domains. But for small molecule drug discovery, SMILES sequences tend not to be extremely long, often 100 tokens or fewer, given the desired molecule weight of the druglike mols ($250 < MW < 500$). Therefore, it is not clear to me why S4 trained on ChEMBL outperforms all the other methods in almost all the tasks. It would be very interesting to conduct a study to investigate how the length of SMILES strings in the training dataset affect the model performance. I would suggest checking the distribution of length in the ChEMBL; if it is not long, authors could try to consider the dataset used in large-scale tasks in Ref [1]. They collect a set of largest molecule (~300k) with more than 100 heavy atoms from PubChem.

2. While some studies have claimed that the language model outperforms graph-based models in a variety of tasks, I think it's still necessary to compare with the graph-based models in long-length SMILES settings. It's impractical to have many graph-based models, but as a methodology paper, it is essential to perform a rigorous study of benchmarking one or two prominent graph-based methods with S4 methods, like GarphVAE/ JTVAE/ HierVAE.

3. Again, as a method paper, I would suggest authors consider more metrics to assess their model (see Ref1) like QED, SA score, Log P, molecular weight, or Bertz complexity, natural product likeness, not just validity, uniqueness, and novelty.

4. I would be concerned about using edit distance as a metric to evaluate the model. Randomized

SMILES string [Ref 2] is a kind of data augmentation method for improving CLM, and there are many ways to draw a molecule in SMILES. I would suggest using 2D similarity instead of edit distance.

5. For bioactivity analysis, I am curious about the similarity between the active compounds used in the training and testing data. How dissimilar are they? If they are similar or coming from the same series, enrichment would not reflect the model's ability to capture bioactivity much. Also, what are the stopping criteria for these fine-tuning tasks? Is the loss associated with these enrichment factors during the fine-tuning?

6. For Fig4b, how was the scaffold clustering performed? The authors need to describe it clearly.

Reference

1. Flam-Shepherd, Daniel, Kevin Zhu, and Alán Aspuru-Guzik. "Language models can learn complex molecular distributions." *Nature Communications* 13.1 (2022): 3293.
2. Arús-Pous, Josep, et al. "Randomized SMILES strings improve the quality of molecular generative models." *Journal of cheminformatics* 11.1 (2019): 1-13.

Reviewer 1

Authors have developed "Structured State-Space Sequence Models for De Novo Drug Design." However, the Manuscript Authors should add the following points to the Manuscript. The Manuscript started with the "Structured State-Space Sequence" Model (S4 model.). The authors have chosen the S4 model for drug design. However, It needs to understand the S4 model. Why have they chosen the model? Any special reason?

Therefore, the authors should discuss the S4 model more clearly.

>> Thank you for this comment. In the revised version of the introduction, we have expanded upon our reasoning to motivate the application of S4 in de novo molecular design. Moreover, we have expanded our analysis to further test the potential of S4 when it comes to SMILES design, in terms of metrics and descriptors, datasets, and molecular dynamics.

In the discussion section, authors should discuss the conventional method of drug discovery. The S4 model is also better than the conventional model. The overall workflow or methodology should be depicted for a clearer view of the readers.

>> In the revised manuscript, we have expanded our considerations on the chosen conventional benchmarks and the reasoning behind their choice (introduction, results and discussion) and on how S4 extends upon them (Section 2). Finally, the revised Figure 1c clarifies the workflow.

Reviewer 2

This manuscript proposes an application of Structured State-Space Sequence (S4) generative models to de novo molecule design. The research aim is significant, and this is the first application of S4 to de novo drug design as claimed (as far as I'm aware);

>> Thank you!

However, several critical issues exist regarding how the computational method is technically explained, the performance is evaluated, the results are discussed, and the source code and trained models are shared with the research community. Unfortunately, this study is far from being a complete work. I'm sorry to state this, but the manuscript strongly feels like it was rushed for publication before it was finished (maybe to be the first to apply S4 to de novo drug design).

>> We appreciate the reviewer's insights and have addressed all their suggestions; please see our point-by-point responses below.

Below, I list my specific concerns/issues:

1) The literature review is not sufficiently detailed regarding generative AI applications in de novo drug design. Generative AI for small molecule and protein design has become highly popular lately. Also, deep learning architectures such as graph neural networks, GANs and VAEs have widely been utilised for this purpose. These studies should be cited and mentioned, especially in the context of deep learning architectures and produced output.

>> We have now (a) expanded our considerations about de novo design, (b) incorporated references to generative models and graph-based molecule generation, as suggested. Finally, we have better motivated the scope of our research and focused on chemical language models. This change is reflected throughout the manuscript and in the title (which was changed into 'Chemical Language Modeling with Structured State Spaces).

2) "2.2 S4 for de novo molecule design"

These tests are not sufficient as a state-of-the-art/baseline comparison. First of all, only validity, uniqueness and novelty are used as performance metrics, which only presents a limited perspective. There are numerous additional metrics to evaluate models from other angles. These metrics should be utilised for evaluation.

>> We fully agree on the limitations of validity, uniqueness, and novelty, when considered alone. In the revised version of the manuscript, we extended our evaluation and considerations as follows:

1. We added explicit considerations on the limitations of validity, uniqueness, and novelty: "Although these metrics are vulnerable to trivial baselines [28], they provide insights into a model's capacity to learn the SMILES 'syntax'..." (lines 161-163)
2. We added a thorough analysis of what factors cause SMILES invalidity for all methods, adding insights to the observed validity (Section 3.1 and Fig. 2).
3. We added five molecular descriptors commonly used in the field (molecular weight, logP, QED, Bertz complexity index, and synthetic accessibility) to provide additional evaluation (lines 166-170 and Supp. Info.).

Moreover, we expanded the breadth of our evaluation, by:

1. Running experiments on natural products with SMILES strings longer than 100 tokens, which shows the potential of S4 to design bigger and more complex molecular entities (Section 3.2).
2. Performing a prospective study supported by molecular dynamics simulations (Sections 3.3).

We believe that the new insights gathered with these analyses support the introduction of S4 as a novel chemical language model with high potential.

Second, the authors only compared their model with LSTM and GPT, which are trained by the authors as their own baselines. However, there are many well-known methods/models in the literature for de novo molecule generation, including both CLMs and graph learning-based models. These models should be employed for performance comparison.

>> We agree that a wealth of approaches exist. Please see our responses below.

1. On the choice of models. The choice of LSTMs and transformers is based on (a) their being the de facto approaches for chemical language modeling (Chen et al. 2023, *Briefings in Functional Genomics* 22, 4), (b) existing scientific evidence on their performance (e.g., Flam-Shepherd et al, 2022. *Nature Communications* 13, 3293; Bagal et al. 2021. *Journal of Chemical Information Modeling* 62, 9, 2064-2076,) and (c) their architectural characteristics compared to S4 (recurrence vs holistic training). These aspects were further emphasized in the manuscript. Moreover, to incorporate the suggestion of the reviewer, we:
 - a. Experimented with NP-VAE (Ochiai et al. 2023, *Communications Chemistry* 6, 249) and HierVAE (Jin, Barzilay, Jaakkola 2020, *ICML*) models. However, they have raised out-of-memory errors for the large molecules in our study (configuration: NVIDIA 40GB GPU, batch size 1, and number of model parameters ~70K).
 - b. As a consequence of point (a), we have expanded our considerations on the choice of CLMs vs graph-based models, according to limitations observed in literature (e.g., Ochiai et al. 2023, *Communications Chemistry* 6, 249; lines 24-27).
 - c. Refocused the scopes of the paper (also in light of point (a)) to chemical language modeling, by making this more explicitly in the title and throughout the manuscript.
2. On the choice of metrics. Here, we considered best practices in the field, e.g., choice of ChEMBL for training, and metrics and descriptors (which were expanded) (Flam-Shepherd et al, 2022. *Nature Communications* 13, 3293; Ochiai et al, 2023. *Communications Chemistry* 6.1: 249). Finally, and more importantly, the tasks were devised to challenge all models equally; here, training all the models with the same hyperparameter tuning pipeline and on the same tasks allowed us to be as fair as possible. We consider this procedure fairer and more transparent than trying to beat pre-existing benchmarks with S4. This aspect is even more important in light of the new analysis using natural products (Section 3.2), where established benchmarks do not exist yet.

Finally, the performance should be discussed with regard to the architecture-, data- and modelling-related differences between the S4 model and the others.

>> We added discussions on the similarities and differences between the architectures in terms of training and generation, with a particular emphasis on the effect of holistic (transformer) and recurrent (LSTM) training, and the combination of both (S4).

3) "2.3 S4 for capturing bioactivity"

It is not clear how this procedure was handled to accommodate the fine-tuning procedure in the context of transfer learning and the production of likelihoods for the test molecules. Technical methodology-related details should be provided. Also, a similar problem exists here (as in issue number 2). Comparison with LSTM and GPT is not sufficient.

>> In the revised version of the manuscript, we (a) simplified our evaluation procedure to increase transparency and added clarifications and methodology-related details on the overall pipeline (Section 5.1); (b) extended the number of repeats performed, to ensure robustness and increase the number of experiments used to compare S4 with LSTM and GPT (c) performed statistical tests to further support the conclusions.

4) "Figure 2 and Figure 3"

While the results provided here seem partially promising, they are far from being conclusive. First of all, baselines performed better in some of the targets. There is no discussion at all regarding this. Additionally, some of these results are contradicting. S4 performed the best for mTORC1 in Figure 2, it performed clearly worse compared to LSTM for mTORC1 in Figure 3; why? Please discuss your results...

>> The experimental setup was extended, by repeating fine-tuning 10 times with different data splits for more consistent results. Moreover, we (a) added statistical tests to improve the conclusiveness of the results, (b) improved our discussion on the differences between methods, and (c) related the results to how "difficult" the datasets are, in terms of similarity between active and inactive molecules (Supporting Information). Thank you for the suggestion.

5) "On GBA and MTORC1, S4 outperforms GPT by 5 to 9 folds enrichment, and LSTMs by more than 17 folds." I cannot see this much difference in Figure 2. Maybe I'm missing something; additional explanation is required.

>> In the revised version of the manuscript, we (a) expanded the explanation regarding the overall procedure, and (b) replaced the enrichment factor (measured in folds compared to random draws) with the fraction of retrieved bioactive hits (expressed in percentage), to ease the interpretation of our results. We believe that this new version will provide more clarity and transparency to the readers.

6) "Interestingly, unlike with bioactivity, in this case LSTM outperforms GPT, on both the validity and the structural diversity with increasing the temperature values (Figure 4a,b). S4 shows an intermediate behavior between LSTM and GPT." This sounds like clever wording, not to admit that LSTM clearly outperformed S4 in terms of the model performance in terms of generation performance with varying temperature values (Figure 4a and b). Please, do not go around the obvious; just say it. Again, there is no discussion at all.

>> This was not our intention. In the new version, we carefully revised our paper to make sure that the cases in which LSTM and/or GPT outperform S4 are as clearly stated as possible. Finally, we have broadened our discussions about the differences in performance between methods and related it as much as possible to differences in their core functioning.

7) "2.5 Generation speed of S4"

Again, the fact that LSTM outperformed S4 is ignored here. By looking at the results provided in this manuscript, I think the overall discussion (considering all test results provided in 2.2, 2.3, 2.4 and 2.5) should be that LSTM performed better than (or at least nearly as good as) S4 in many of the tests (except maybe some of the results in Figure 2 and 3), which indicates

LSTM, an old but effective architecture, is as promising as the S4 model in de novo molecule design until further analyses indicate otherwise.

>> The revised version contains an array of additional analyses of the differences between methods. We believe that our results clearly show that S4 and LSTM have distinct advantages. S4 has an edge in designing complex SMILES, and at capturing the properties of the training set (e.g., bioactivity and NP-likeness). LSTMs is slightly better in designing novel molecules, but it performs worse than S4 in designing molecules with bespoke properties. Hence, we do believe that S4 is a valuable addition to the family of chemical language models and that it can complement and/or outperform LSTMs. These aspects have been further stressed in our discussion.

8) It is strange that not one de novo molecule is shown in a study in which a de novo molecule design model is proposed. Sample molecules from each test should be given, and models should be compared to each other over the generated molecules. This can be shaped in the context of a use case study, in which a few example de novo molecules generated by the S4 model are qualitatively compared with the output of other models from physicochemical and functional perspectives.

>> We agree, thank you. The revised version of the manuscript was extended to:

1. Compare the properties of the de novo designs quantitatively, using an array of molecular descriptors and properties, both related to drug-like molecules and natural products.
2. Include a prospective in-silico study (Section 3.3) corroborated by extensive molecular dynamics. Here, we show and analyze the de novo designs in depth, in a quantitative (predicted binding affinity) and qualitative manner (e.g., Figure 5).

9) One of the most critical issues is the missing technical explanation of the architecture in the methods section. Authors correctly claim that S4 models are first used here for de novo molecule generation, but they did not explain the details of the architecture. I'm aware that the authors did not propose the S4 architecture itself; it has been previously developed by others, but a thorough technical explanation is required in a study where a rather new architecture is applied in another field for the first time.

>> We have now surfaced the explanation of S4 by moving it to a separate section and adding more details. We have omitted a full in-depth explanation to avoid redundancy with the original paper (Gu et al. 2022) and keep the paper accessible to a broad readership.

10) An ablation study is required to evaluate the contribution of different modules of the proposed S4-based model to the produced output. This could be done by eliminating certain modules or replacing them with more conventional counterparts. The change in the results should be discussed in relation to the original design.

>> The paper that proposed S4 (Gu et al. 2022) contains ablation studies. The results of these ablation studies are now briefly mentioned in the revised version of the manuscript (Section 2, lines 110-113).

11) The authors just stated that the code will be shared over the given GitHub repo link (the repo is empty as of the last week of October 2023). In today's research, open science is one of the most important aspects, and sharing of the scripts/source code is an important part of it. The source code, datasets and results should be properly shared before the peer review process, not after (so that the reproducibility could be checked). On top of sharing the source code, please prepare a detailed readme file that includes information regarding how your

implementation can be used to reproduce the results provided in the manuscript. In addition, please provide information on how the implementation can be used to analyze independent datasets belonging to the user. This way, future researchers can use your method in their own research, and they also can replicate your results when and where necessary.

>> The code, documentation, and usage guidelines were published while the paper was under review, and this Reviewer missed it for (most likely) a few days. In fact, Reviewer 3 managed to access it and complimented the code for its style and clarity.

Reviewer 3

This manuscript by Riza et al. introduces a new generative model for de novo generation of molecules from the control engineering research. This is the first time to apply the structured state-space sequence (S4) model (existing approach, SOAT in other domains) to drug design. Authors benchmarked the methods with the current two SOTA methods (LSTM and GPT) on various small molecule drug discovery-related tasks and showed it outperforms all the other methods. The methodology is described clearly, and the code in GitHub is well-documented and maintained.

>> Thank you for appreciating our work, and our GitHub repository.

However, I have serious reservations regarding the value of the proposed approach for drug discovery, especially on small molecule drug design, based on the method and results.

1. The most important issue is the motivation for applying S4 to SMILES-based small molecule generation. S4 was proposed to deal with very long sequences, more than 10000 or more steps, and it has been proven to be effective in many domains. But for small molecule drug discovery, SMILES sequences tend not to be extremely long, often 100 tokens or fewer, given the desired molecule weight of the druglike mols ($250 < MW < 500$). Therefore, it is not clear to me why S4 trained on ChEMBL outperforms all the other methods in almost all the tasks.

>> Indeed, S4 was shown to 'shine' with longer sequences, but this is not all. It is known that LSTMs and GPT have different advantages, due to their distinct natures (recurrence vs holistic training and generation). These differences are reflected in their potential for chemical language modeling (e.g., Chen et al. 2023, *Briefings in Functional Genomics* 22, 4) in terms of the properties they learn and their applicability to molecule design. S4 (combining recurrence and holistic training) has the potential to 'make the best of both worlds' and advance the current state-of-the-art – and this is the starting point of our research, and our main conclusion. These aspects have now been clarified in the revised version of the manuscript.

It would be very interesting to conduct a study to investigate how the length of SMILES strings in the training dataset affect the model performance. I would suggest checking the distribution of length in the ChEMBL; if it is not long, authors could try to consider the dataset used in large-scale tasks in Ref [1]. They collect a set of largest molecules (~300k) with more than 100 heavy atoms from PubChem.

>> The revised version was now expanded to include longer SMILES sequences, and particularly those of natural products, which are not only longer, but contain more long-range dependencies. To this end, we used the COCONUT database (30,000 SMILES between 100 and 450 characters). The new results show that S4 can better learn to design molecules matching the complex properties of the natural products used for training. Thank you for your suggestion.

2. While some studies have claimed that the language model outperforms graph-based models in a variety of tasks, I think it's still necessary to compare with the graph-based models in long-length SMILES settings. It's impractical to have many graph-based models, but as a methodology paper, it is essential to perform a rigorous study of benchmarking one or two prominent graph-based methods with S4 methods, like GraphVAE/ JTVAE/ HierVAE.

>> We have experimented with HierVAE and NP-VAE since they were the most promising methods for the size of the molecules in this study. However, both models resulted in out-of-memory errors despite decent hardware configuration and small models (NVIDIA 40GB GPU, batch size 1, number of model parameters ~70K). A similar problem is observed in the NP-VAE paper itself (Ochiai et al, 2023. *Communications Chemistry* 6.1: 249). After this evidence, we now honed the scope of our manuscript to chemical language modeling, as reflected throughout the manuscript and in the new title.

3. Again, as a method paper, I would suggest authors consider more metrics to assess their model (see Ref 1) like QED, SA score, Log P, molecular weight, or Bertz complexity, natural product likeness, not just validity, uniqueness, and novelty.

>> The revised version of the manuscript now includes logP, molecular weight, QED and Bertz complexity for ChEMBL (Supporting Information), and NP-likeness alongside other descriptors for COCONUT (Section 3.2). The first version already contained both SA and SC scores (Supporting Information).

4. I would be concerned about using edit distance as a metric to evaluate the model. Randomized SMILES string [Ref 2] is a kind of data augmentation method for improving CLM, and there are many ways to draw a molecule in SMILES. I would suggest using 2D similarity instead of edit distance.

>> We agree. The Edit distance was replaced with Tanimoto similarity computed on extended connectivity fingerprints, which is invariant to atom ordering. The analysis was repeated.

5. For bioactivity analysis, I am curious about the similarity between the active compounds used in the training and testing data. How dissimilar are they? If they are similar or coming from the same series, enrichment would not reflect the model's ability to capture bioactivity much.

>> We use random splitting to create train/test splits and repeat the splitting 10 times. We plotted the train-test similarity (Figure S4) and discovered new insights about the performance of the chemical language models. We discuss those in Section 3.1.2 (lines 255-258). Thank you for the suggestion.

Also, what are the stopping criteria for these fine-tuning tasks? Is the loss associated with these enrichment factors during the fine-tuning?

>> We did not use the enrichment for stopping the training, to simulate a real-world scenario where information on bioactivity would not be accessible. We use validation loss (cross-entropy) for early stopping. These details are now reported in the Materials and Methods section (lines 600-602).

6. For Fig4b, how was the scaffold clustering performed? The authors need to describe it clearly.

>> This is now elaborated in Section 3.1.3. We used hierarchical clustering with Tanimoto similarity on scaffold Morgan Fingerprints and ensured that molecules with scaffold similarity over 60% are placed in the same cluster.

REVIEWER COMMENTS

Reviewer #2 (Remarks to the Author):

The authors improved their manuscript by addressing many of the reviewers' comments. I appreciate the newly added analyses. However, there are still some critical concerns. I will mention them using the issue numbers from the previous round of review (considering reviewer number 2) and two new ones.

1)

The focus has been shifted to CLMs. However, the recent CLM literature is not cited. There are many recent CLM studies (I mean original research) that came out in 2021, 2022 and 2023; which are extremely relevant to the proposed work (nearly all of them utilised SMILES or SELFIES representation-based transformers); however, most of them are not cited. The authors heavily cited survey/review papers instead. This should be fixed.

2)

New metrics have been added, but they only remain in Figure S1, which is nearly impossible to interpret because distributions for 5 molecular descriptors are given side by side without any discussion or insight anywhere in the main text or supplementary material. Even the mean values have not been provided.

Also, as part of issue #2, comparing with LSTM and GPT is well justified, and I agree; however, this is a baseline comparison, as the authors also mentioned (since S4 improved over these architectures), and not a state-of-the-art comparison, which is still missing from the manuscript. Even though authors choose to focus on CLMs (and not graph-based models) there are many recent CLMs in the literature, so they could just compare their performance with those. Without a real state-of-the-art comparison, the study is incomplete. Those CLMs are also mostly based on transformers, but unlike the one trained-tested by the authors of the proposed study, those CLMs have special modules/tweaks that put them apart from vanilla versions of their fundamental architectures. That is why it is important to compare with real, previously developed and published methods from the literature, apart from the baseline model comparison analysis in which you create/train/test your own baseline models.

If testing state-of-the-art models is not possible on all the existing tasks, well-known benchmarks (e.g., molecular property prediction) can be utilized for this comparison.

11)

The authors have commendably presented the code of their model in a clear and accessible format, which is a positive aspect of their work. However, omitting crucial elements such as the weights of the models trained and the generated de novo molecules used in analyses poses significant challenges for replicating and validating their findings. Without access to these critical components, external researchers cannot verify the results or assess the model's performance accurately without large-scale training. Sharing the model weights and the details of the molecules used in the analyses would greatly enhance the utility and credibility of their contributions, facilitating further advancements in the field. Please share these missing bits.

A general critical concern about the study:

Maybe the most critical issue still is about the advantages of S4 brought to the field chemical language modelling. I'll try to explain below with quotes from the manuscript.

(abstract) "S4 has a remarkable capability to capture the global properties of long sequences. This aspect is key for chemical language modeling, where complex molecular properties like bioactivity can 'emerge' from distant positions in the molecular strings." and

(lines 44-46) "Transformers computationally intensive for string generation, thereby potentially limiting their chemical space exploration capabilities."

While the authors emphasize S4's remarkable ability to capture global properties of long sequences, suggesting its potential utility in modelling complex molecular properties that emerge from distant positions in molecular strings, it also points out the computational intensiveness of Transformers in generating strings, which could limit their exploration of chemical space. However, it is important to note that these claims may not be entirely relevant to the domain of small molecules (including natural products). Typically, small molecules are represented by sequences with a maximum length of around 100 characters, a scale at which Transformer models are known to operate efficiently. Furthermore, S4 models are specifically designed/used to handle very long sequences, often exceeding lengths of 10,000, far beyond the typical sequence length encountered in small molecule / natural product representations. Therefore, the discussion on the suitability and efficiency of S4 and Transformer models in the paper seems to overlook that the sequence lengths commonly associated with chemical molecules do not necessitate the advanced capabilities of S4 models designed for much longer sequences. This oversight calls into question the applicability of the paper's claims to real-world chemical language modelling, suggesting a need for a more nuanced understanding of the requirements of this domain.

Conducted analyses put S4 forward in some tests and GPT/LSTM for others. So, there is no consensus here. Would it be possible to show a clear advantage, considering the domain of small molecule / natural product learning and design, in relation to the theoretical advantages of state space models?

Concerns about newly conducted analyses:

A)

About "3.2 Designing natural products", why is S4 not faster than GPT in training? Isn't this one of the most critical advantages of state-space architectures?

B)

About "3.3 Prospective de novo design", Tanimoto similarities between de novo designs and the most similar training MAPK1 inhibitors should also be provided and the discussion should include this information.

Reviewer #2 (Remarks on code availability):

The authors have commendably presented the code of their model in a clear and accessible format, which is a positive aspect of their work. However, omitting crucial elements such as the weights of the models trained and the generated de novo molecules used in analyses poses significant challenges for replicating and validating their findings. Without access to these critical components, external researchers cannot verify the results or assess the model's performance accurately without large-scale training. Sharing the model weights and the details of the molecules used in the analyses would greatly enhance the utility and credibility of their contributions, facilitating further advancements in the field. Please share these missing bits.

Reviewer #3 (Remarks to the Author):

The Authors have addressed all of my concerns with the revised manuscript. I support this manuscript for publication.

Reviewer 2

The authors improved their manuscript by addressing many of the reviewers' comments. I appreciate the newly added analyses. However, there are still some critical concerns. I will mention them using the issue numbers from the previous round of review (considering reviewer number 2) and two new ones.

>> Thank you for the positive feedback on the analyses we added. In what follows, please find our point-by-point response to the remaining comments.

1) The focus has been shifted to CLMs. However, the recent CLM literature is not cited. There are many recent CLM studies (I mean original research) that came out in 2021, 2022 and 2023; which are extremely relevant to the proposed work (nearly all of them utilised SMILES or SELFIES representation-based transformers); however, most of them are not cited. The authors heavily cited survey/review papers instead. This should be fixed.

>> Done, thank you for the suggestion. Several recent papers have been referenced in the revised version of the manuscript, including transformer-based architectures and other CLM literature (L13):

1. Hong et al. 2022. Molecule Generation for Drug Discovery with New Transformers. SSRN. DOI: 10.2139/ssrn.4195528.
2. Wang et al. 2023. cMolGPT: A conditional generative pre-trained transformer for target-specific de novo molecular generation. *Molecules* **28**, 4430. DOI: 10.3390/molecules28114430.
3. He et al. 2024. TD-GPT: Target protein-specific drug molecule generation GPT. *IEEE International Conference on Acoustics, Speech and Signal Processing (2024)*. DOI: 10.1109/icassp48485.2024.10447303.
4. Hu et al. 2024. De novo drug design using reinforcement learning with multiple GPT agents. *Advances in Neural Information Processing Systems (NeurIPS)*. DOI: 10.48550/arXiv.2401.06155.
5. Gummesson Svensson et al. 2024. Utilizing reinforcement learning for de novo drug design. *Machine Learning* **1**, 33 (2024). DOI: 10.1007/s10994-024-06519-w.

2) New metrics have been added, but they only remain in Figure S1, which is nearly impossible to interpret because distributions for 5 molecular descriptors are given side by side without any discussion or insight anywhere in the main text or supplementary material. Even the mean values have not been provided.

>> Done. The supporting material has been enriched with summary statistics on these properties to improve interpretability (Table S1). The considerations on this information in the main text have also been further expanded (L171-177).

Also, as part of issue #2, comparing with LSTM and GPT is well justified, and I agree; however, this is a baseline comparison, as the authors also mentioned (since S4 improved over these architectures), and not a state-of-the-art comparison, which is still missing from the manuscript. Even though authors choose to focus on CLMs (and not graph-based models) there are many recent CLMs in the literature, so they could just compare their performance with those. Without a real state-of-the-art comparison, the study is incomplete. Those CLMs are also mostly based on transformers, but unlike the one trained-tested by the authors of the proposed study, those CLMs have special modules/tweaks that put them apart from vanilla versions of their fundamental architectures. That is why it is important to

compare with real, previously developed and published methods from the literature, apart from the baseline model comparison analysis in which you create/train/test your own baseline models. If testing state-of-the-art models is not possible on all the existing tasks, well-known benchmarks (e.g., molecular property prediction) can be utilized for this comparison.

>> We partially agree with the reviewer's opinion. While we recognize the relevance of benchmarks in the AI landscape, it is also true that:

- Comparing vanilla versions (with no 'extra tweaks') provides more authentic insights into the potential of a given architecture and its functioning. Tweaks, in fact, can be used to achieve a desired, and task-specific performance (e.g., beating a certain benchmark), and do not necessarily reflect the broad applicability/potential in a field.
- In this study, we use S4 in a 'vanilla' version, which puts it at a disadvantage compared to 'tweaked' models based on well-established architectures (which have undergone extensive prior experimentation). This is why we have compared S4 with other architectures in their vanilla form.
- Benchmarks are currently under scrutiny in the AI community (e.g., Raji et al. 2021. AI and the 'everything in the whole wide world' benchmark. *arXiv preprint*, arXiv:2111.15366). In fact, not only are benchmarks far from being indicative of true general potential, but they might lead to "state-of-the-art chasing" and encourage incremental research.

To comply with the reviewer's request, we have benchmarked S4 with the widely adopted MOSES benchmark for de novo molecule design (Polykovskiy et al. 2020. *Frontiers in Pharmacology* **11**, 565644). Using MOSES, S4 was compared with 11 de-novo design approaches. Eight of them are generative deep learning architectures, including recent transformer-based CLMs and corresponding tweaks (MD-TF, cMolGPT, and TD-GPT), as requested. S4 – in its vanilla version — consistently scored among the top-3 deep learning approaches across metrics, showing the potential of this architecture to be expanded further. These results are now included in Table S2 and mentioned in the manuscript (L186-190).

11) The authors have commendably presented the code of their model in a clear and accessible format, which is a positive aspect of their work. However, omitting crucial elements such as the weights of the models trained and the generated de novo molecules used in analyses poses significant challenges for replicating and validating their findings. Without access to these critical components, external researchers cannot verify the results or assess the model's performance accurately without large-scale training. Sharing the model weights and the details of the molecules used in the analyses would greatly enhance the utility and credibility of their contributions, facilitating further advancements in the field. Please share these missing bits.

>> Done. We thank the reviewer for their suggestion and agree that this information will benefit the community further. We have now released model weights, designs, and log-likelihoods on Zenodo (available at the following DOI: [10.5281/zenodo.11085649](https://doi.org/10.5281/zenodo.11085649)). This material and the corresponding GitHub repository are now cross-linked.

A general critical concern about the study:

Maybe the most critical issue still is about the advantages of S4 brought to the field chemical language modelling. I'll try to explain below with quotes from the manuscript.

(abstract) “S4 has a remarkable capability to capture the global properties of long sequences. This aspect is key for chemical language modeling, where complex molecular properties like bioactivity can ‘emerge’ from distant positions in the molecular strings.” and (lines 44-46) “Transformers computationally intensive for string generation, thereby potentially limiting their chemical space exploration capabilities.”

While the authors emphasize S4’s remarkable ability to capture global properties of long sequences, suggesting its potential utility in modelling complex molecular properties that emerge from distant positions in molecular strings, it also points out the computational intensiveness of Transformers in generating strings, which could limit their exploration of chemical space. However, it is important to note that these claims may not be entirely relevant to the domain of small molecules (including natural products). Typically, small molecules are represented by sequences with a maximum length of around 100 characters, a scale at which Transformer models are known to operate efficiently.

>> We would like to provide further clarity on the broader context of this study:

- On modelling complex molecular properties. Existing literature suggests that LSTMs and Transformers have different and complementary advantages/disadvantages when it comes to generating structurally molecules de novo, and learning local vs global properties (e.g., Chen et al. 2023. *Briefings in Functional Genomics* **22**, 392; Gómez-Bombarelli et al. 2018. *ACS central science* **4**, 268). Here, we hypothesize that the dual formulation of S4 (i.e., similar to Transformers in training, and to LSTMs in generation) can make the ‘best of both worlds’ to advance chemical language modelling. Our results show indeed that S4 (a) outperforms LSTMs and GPTs in their respective weaknesses, and (b) has the best trade-off between exploring novel structures and matching complex molecular properties.
- On computational intensiveness. S4s have a linear algorithmic complexity in generation length, whereas GPTs have a quadratic complexity. While we acknowledge that the SMILES of drug-like molecules are usually in the order of 100 tokens, a typical de novo design project might involve generating millions of molecules. Here, the speed-up provided by S4 is highly desirable. With longer sequences (e.g., natural products, Figure S5) computational efficiency becomes not only desirable, but also strictly necessary. Overall, well-performing methods that are computationally more efficient have higher potential and higher versatility when it comes to their application in the molecular sciences.

To provide further clarity on the aspects discussed above, the two sentences mentioned by the reviewer have been now improved as follows:

- “S4 has shown remarkable capabilities to learn global properties of sequences. This aspect is intriguing in chemical language modeling, where complex molecular properties like bioactivity can ‘emerge’ from separated portions in the molecular string.” (Abstract)
- “Moreover, while LSTMs remain efficient, Transformers become increasingly compute-intensive when generating progressively longer SMILES strings, which might limit their broad applicability in the chemical sciences.” (L61-65)

Finally, we have added the following sentence, to improve our manuscript further:

- “LSTMs and GPTs present different – and somewhat complementary – strengths and weaknesses when it comes to de novo molecule design [25, 29–32]. The ‘recurrent’ nature of LSTMs allows learning local properties better than GPTs, while GPTs capture global properties better thanks to their ‘holistic’ processing [25].” (L55-61)

Furthermore, S4 models are specifically designed/utilized to handle very long sequences, often exceeding lengths of 10,000, far beyond the typical sequence length encountered in small molecule / natural product representations. Therefore, the discussion on the suitability and efficiency of S4 and Transformer models in the paper seems to overlook that the sequence lengths commonly associated with chemical molecules do not necessitate the advanced capabilities of S4 models designed for much longer sequences.

>> The fact that S4 has been designed explicitly to handle very long sequences does not automatically exclude it from working well on shorter sequences in different application domains. When considering the specific case of molecules, in fact, (even short) SMILES strings contain dozens of nested dependencies, where atoms that are close in the molecular graph can be separated by several dozens of characters in the string, and many branches can exist. This makes SMILES sequences a very 'special' case, where considerations from other application domains might not necessarily apply. Contrary to the reviewer's expectations, the systematic results obtained in this study show that S4 is indeed well suited to the molecular domain.

This oversight calls into question the applicability of the paper's claims to real-world chemical language modelling, suggesting a need for a more nuanced understanding of the requirements of this domain.

>> We would like to kindly redirect the reviewer to some of our works on chemical language modelling for drug discovery, such as: Moret et al. (2024) *Nature Communications* **14**, 114; Grisoni et al. (2021) *Science Advances* **7**; Grisoni et al. (2020) *J. Chem. Inf. Mod.* **60**; Özçelik et al. (2021) *Molecular Informatics* **40**, 2000212; Özçelik et al. (2024) *Molecular Informatics* **43**, e202300249; Grisoni (2023) *Current Opinion in Structural Biology* **79**, 102527; Moret et al. (2020) *Nature Machine Intelligence* **2**, 171; Ballarotto et al. (2023). *Journal of Medicinal Chemistry* **66**, 8170.

Conducted analyses put S4 forward in some tests and GPT/LSTM for others. So, there is no consensus here. Would it be possible to show a clear advantage, considering the domain of small molecule / natural product learning and design, in relation to the theoretical advantages of state space models?

>> Our results show that there is a consensus (line 533):

"[...] while recurrent generation (LSTM and S4) is superior in learning the chemical syntax and exploring diverse scaffolds, learning holistically on the entire SMILES sequence (GPT and S4) excels in capturing certain complex properties, like bioactivity. S4 with its dual nature, makes 'the best of both worlds': it demonstrated comparable or better performance than LSTM in designing valid and diverse molecules, and systematically outperformed both benchmarks in capturing complex molecular properties – all while maintaining computational efficiency." Hence, conducting further theoretical analysis is out of scope, and might be the object of future works.

Concerns about newly conducted analyses:

A) About "3.2 Designing natural products", why is S4 not faster than GPT in training? Isn't this one of the most critical advantages of state-space architectures?

>> Our results on computational efficiency are in line with the original S4 theory (see Table 1, in: Gu et al. 2021, Efficiently modeling long sequences with structured state spaces. *arXiv preprint arXiv:2111.00396*. DOI: <https://doi.org/10.48550/arXiv.2111.00396>).

B) About “3.3 Prospective de novo design”, Tanimoto similarities between de novo designs and the most similar training MAPK1 inhibitors should also be provided and the discussion should include this information.

>> Done, thank you for the suggestion. The similarities were added to the revised version of the manuscript, *i.e.*, in Table 3, and the corresponding results were discussed (L478-510).

REVIEWERS' COMMENTS

Reviewer #2 (Remarks to the Author):

I thank the authors for addressing the remaining concerns.